# Stakeholder priorities and conceptualization of One Health: Insights from fuzzy cognitive mapping and grounded theory

Evan F. Griffith[ID][1,2*], Angela Opondoh[ID][2,3], Catherine Kaluwa[4], Erenius Lochede Nakadio[5,6], Kipkorir Rotich[5,7], Job Ronoh Kipkemoi[5,6], Jonah Levin[1], Jacob Mutua[5,8], Siobhan M. Mor[9,10], Janetrix Hellen Amuguni[1]

**1** Cummings School of Veterinary Medicine, Tufts University, North Grafton, Massachusetts, United States of America, **2** NEOH – The Network for Ecohealth and One Health, Deurne, Belgium, **3** Institute of Anthropology, Gender and African Studies, University of Nairobi, Nairobi, Kenya, **4** Department of Veterinary Anatomy and Physiology, University of Nairobi, Nairobi, Kenya, **5** County One Health Unit, Turkana County Government, Lodwar, Kenya, **6** Department of Agriculture, Livestock Development, and Fisheries, Turkana County Government, Lodwar, Kenya, **7** Department of Health Services and Sanitation, Turkana County Government, Lodwar, Kenya, **8** Department of Tourism, Culture, Natural Resources, and Climate Change, Turkana County Government, Lodwar, Kenya, **9** University of Liverpool, Leahurst Campus, Neston, United Kingdom, **10** International Livestock Research Institute (ILRI), Nairobi, Kenya

* evan.griffith@tufts.edu

## Abstract

One Health (OH) has gained global recognition as a framework and practice for addressing interconnected health and sustainability challenges, such as emerging infectious diseases, food insecurity and climate change. Yet its operationalization remains limited, in part due to persistent differences in how OH is conceptualized across sectors and knowledge systems, including Indigenous knowledge holders and environmental actors. To address these gaps, we applied a mixed methods approach that combined fuzzy cognitive mapping (FCM) with grounded theory (GT) to examine stakeholder priorities and conceptualization of OH in Turkana County, Kenya. Thirty-six fuzzy cognitive maps were co-developed with community members, frontline workers, health, veterinary, and NGO stakeholders, then aggregated by group and analyzed using network metrics including normalized degree centrality and Jaccard similarity. Results revealed areas of alignment and convergence in OH conceptualization and perspective. Conceptually, the environment group map was the most distinct, while the health and NGO maps were the most similar. Environmental actors emphasized human-driven degradation, while community members described a pluralistic health system and the importance of wild foods, both absent from other group narratives. Shared priorities such as human and livestock health, nutrition, and water resources represent potential entry points for cross-sectoral integration. These findings demonstrate that barriers to OH operationalization are not only structural, but also shaped by differences in knowledge, experience, and problem framing across sectors and society within the socioecological system. Embedding ecosystem

**Data availability statement:** All relevant data are within the manuscript and its Supporting information files.

**Funding:** This research was generously funded by a Cummings Foundation grant (V710458) to H.J.A. Website: https://www.cummingsfoundation.org/. The funders had no role in study design, data collection and analysis, decision to publish, or preparation of the manuscript.

**Competing interests:** The authors have declared that no competing interests exist.

services, biodiversity, and traditional knowledge at the core of OH can enhance inclusivity and contextual relevance. Our integrated FCM-GT approach offers a transferable framework for participatory systems research, providing actionable insights for more inclusive and ecologically grounded OH implementation.

## Introduction

Over the past few decades, One Health (OH) has gained widespread global recognition as a conceptual framework and practice for addressing interconnected threats to human, other animal, and environmental health [1]. Yet despite its prominence and broad conceptual appeal, operationalizing OH remains difficult, hindered by technical, institutional, and structural barriers, as well as issues of sustainability and competing priorities [2–4]. These limitations have hindered efforts to design integrated policies and generate actionable data across complex socioecological systems (SES), preventing OH from realizing its full potential [5].

A key factor underlying these limitations is the underrepresentation of environment actors in OH policy and practice. While OH reportedly recognizes human, animal, environment interconnectedness [6], environmental stakeholders, such as those involved in natural resource management, water and land governance, and biodiversity conservation, are often excluded from OH policies and initiatives [7–9]. This exclusion limits the ability of OH initiatives to address the ecological determinants of health and undermines efforts to build cross-sectoral responses to complex, interconnected challenges [5].

Traditional knowledge systems are also underrepresented, even though Indigenous ways of knowing and acting have long recognized and practiced interconnections between people, other animals, and the environment [10–12]. Such knowledge is especially important given its long-term, place-based understanding of ecological dynamics and their intersection with human and other animal relationships, which is critical to achieving OH goals, particularly incorporating environmental dimensions [13–15]. In addition, Indigenous knowledge systems offer not just ecological insights, but also governance models and ethical frameworks that challenge dominant Western paradigms [16]. The perspectives of traditional knowledge holders must therefore be included to support transdisciplinary collaboration and strengthen the design and implementation of OH tools and initiatives [6].

Despite their relevance, perspectives of traditional knowledge holders and environmental actors, have rarely been engaged into analyses of how OH itself is conceptualized, leaving a critical gap in understanding how different actors define priorities and perceive system linkages [3]. This is a significant oversight, as stakeholder understanding and perceptions of OH shapes prioritization, sector engagement, and the feasibility of OH interventions [17]. Understanding how different groups conceptualize OH can help to identify areas of convergence that facilitate joint objectives, while also revealing points of divergence that require negotiation or targeted interventions.

## Methods

### Study area

This study was conducted in Turkana County, the largest county in Kenya with a total area of 77,000 km² (Fig 1). The county has an estimated population of 1,256,152 people, the majority of whom are pastoralists [18]. Other livelihoods in the county include fishers, artisanal mining, and urban and peri-urban wage labor [19]. Livestock populations are estimated at 2,828,010 cattle, 6,731,414 sheep, 6,906,686 goats, 871,707 camels, and 623,312 donkeys [19].

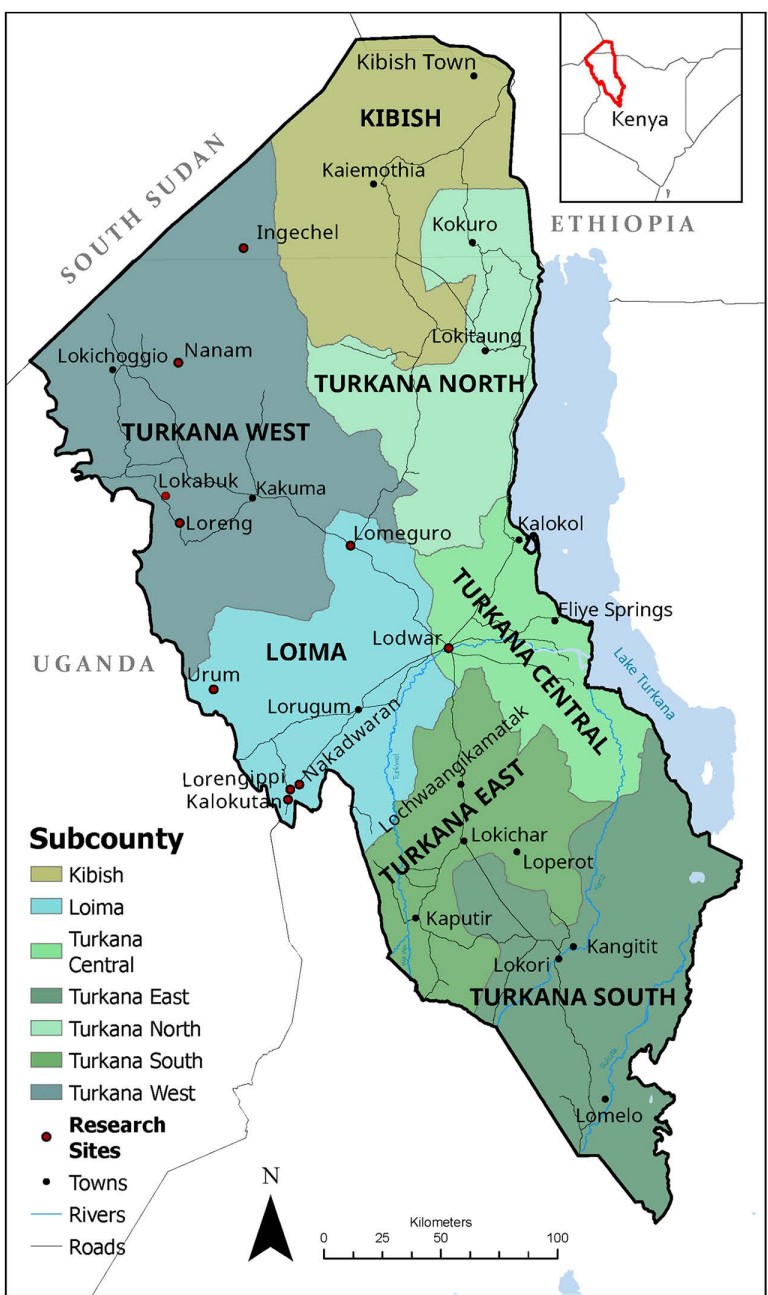

**Fig 1. Map of Turkana County, Kenya.** Research sites include group and individual mapping sessions. Created in Arcmap GIS (10.8.2).

The climate in Turkana is characterized as arid and semi-arid (ASAL) with mean annual rainfall ranging from 400–600 mm. Rainfall follows an east-west gradient, with the long rains (*akiporo*) falling from April to July, and the dry season (*akamu*) from January to February. Vegetation cover is shaped by rainfall and soil characteristics and includes grasses and herbaceous plants, dwarf shrubs and brushland, and riverine habitat with socio-ecologically important species such as the umbrella acacia (*Acacia tortilis*), doum palm (*Hyphaene coriacea*), and the toothbrush tree (*Salvador persica*) [20,21]. The invasive tree, *Prosopis julifloris* (*etirae*), is increasingly common across Turkana, displacing native vegetation and undermining ecosystem services that support pastoral livelihoods [22–24].

Since devolution, service delivery has been overseen by the Turkana County Government in collaboration with the national government, international and national non-governmental organizations (NGOs) [25]. Health and development indicators remain significantly poorer than national averages. For example, 75% of the population fall within the lowest wealth quintile and the under-five mortality rate is 65 per 1,000 live births compared to 41 per 1,000 nationally [26]. The global acute malnutrition rate remains above 15% as of 2024 – at an emergency level [27].

OH activities are coordinated through the County One Health Unit (COHU), which is co-chaired by the directors of the veterinary, health, and environment directorates. COHU also includes designated focal persons from each department [28].

## Sampling and participant recruitment

We purposively sampled OH stakeholders across Turkana County, including community members; community-based actors and representatives from the government departments involved in human health, animal health, and environment; and NGOs (Table 1). We conducted both group and individual FCM mapping sessions, aiming for gender parity across participants.

Participant recruitment occurred from 15/01/2024 to 23/02/2024. Eligibility criteria included being ≥18 years of age and employed in a government department or NGO that has expertise in OH-related domains, a recognized community leadership role, and/or a community member. Initial participants were recruited through COHU as part of an ongoing Kimormor initative – an integrated OH service delivery initiative [29,30]. The research team traveled with the Kimormor team and, with the assistance of subcounty officials, local administrators, and community leaders, recruited participants at each Kimormor site according to the inclusion criteria. Subsequent recruitment in Loima subcounty relied on local administrators and community leaders to help identify suitable participants. We sought diversity in age, gender, geography, and livelihood background.

## Data collection and analysis

Data collection occurred from 29/01/2024 to 23/02/2024. Individual mapping activities with key informants were conducted in English or Kiswahili. Group mapping activities were led in English and Kiswahili, with real-time translation to and from *Nga'turkana* by trained local interpreters. Each group and individual mapping session lasted approximately an hour and was audio-recorded for accuracy and transcribed for analysis.

**Fuzzy cognitive mapping.** FCM is a participatory systems modeling approach that represents participant knowledge as a directed, weighted network of concepts and causal relationships. Concepts (nodes) correspond to elements of the system, while edges represent the direction and strength of perceived causal influence between concepts. Positive edges indicate a direct relationship (i.e., components move in the same direction), while negative edges indicate an indirect relationship (i.e., components move in opposite directions). Each edge has a weight ranging from 0–1, with values closer to 1 indicating a stronger influence. Fuzzy cognitive maps can be represented as adjacency matrices, enabling mathematical aggregation of individual maps into group maps [31].

**Table 1. Participant characteristics, including location, gender makeup, and stakeholder group. Community frontline workers (CFWs) include community health promoters (CHPs) and community animal disease reporters (CADRs).**

| Mapping Session | Location | | Gender | | Stakeholder Group (notes) |
|---|---|---|---|---|---|
| | *Subcounty* | *Town/village/kraal* | *Women* | *Men* | |
| 1 | Turkana Central | Lodwar | 0 | 3 | N/A (Pilot) |
| 2 | Turkana West | Lokabuk Kraal | 0 | 8 | Community |
| 3 | Turkana West | Lokabuk Kraal | 8 | 0 | Community |
| 4 | Turkana West | Loreng Village | 3 | 5 | CFWs (CHPs + CADRs) |
| 5 | Turkana North | Lomeguro Kraal | 4 | 4 | CFWs (CADRs only) |
| 6 | Turkana North | Lomeguro Kraal | 4 | 2 | Environment |
| 7 | Turkana North | Lomeguro Kraal | 4 | 4 | CFWs (CHPs only; No map produced due to time constraints) |
| 8 | Lokichogio | Ingechel Kraal | 0 | 8 | Community |
| 9 | Lokichogio | Nanam Village | 9 | 0 | Community |
| 10 | Lokichogio | Nanam Village | 0 | 8 | Community |
| 11 | Lokichogio | Nanam Village | 8 | 0 | Community |
| 12 | Lokichogio | Nanam Village | 0 | 8 | Community |
| 13 | Loima | Urum Kraal | 0 | 8 | Community |
| 14 | Loima | Urum Kraal | 0 | 8 | Community |
| 15 | Loima | Urum Kraal | 6 | 0 | Community |
| 16 | Loima | Urum Kraal | 8 | 0 | Community |
| 17 | Loima | Nakadwaran Kraal | 8 | 0 | Community |
| 18 | Loima | Nakadwaran Kraal | 0 | 8 | Community |
| 19 | Loima | Kalokutan Kraal | 0 | 8 | Community |
| 20 | Loima | Kalokutan Kraal | 8 | 0 | Community |
| 21 | Turkana West | Loreng Village | 0 | 1 | Veterinary |
| 22 | Turkana West | Lokabuk Kraal | 0 | 1 | Administrator |
| 23 | Turkana North | Lomeguro Kraal | 0 | 1 | Health |
| 24 | Turkana North | Lomeguro Kraal | 0 | 1 | Administration |
| 25 | Lokichogio | Ingechel Kraal | 0 | 1 | Health |
| 26 | Lokichogio | Ingechel Kraal | 0 | 1 | Administration |
| 27 | Lokichogio | Nanam Village | 1 | 0 | Environment |
| 28 | Turkana Central | Lodwar | 0 | 2 | Environment |
| 29 | Turkana Central | Lodwar | 0 | 1 | Environment |
| 30 | Turkana Central | Lodwar | 0 | 1 | NGO (Turkana Pastoralist Development Organization) |
| 31 | Turkana Central | Lodwar | 0 | 1 | NGO (International Rescue Committee) |
| 32 | Turkana Central | Lodwar | 0 | 1 | NGO (USAID Imarisha Jamii) |
| 33 | Turkana Central | Lodwar | 1 | 0 | NGO (USAID Nawiri) |
| 34 | Turkana Central | Lodwar | 0 | 1 | Administration (Office of the Governor Official) |
| 35 | Loima | Lorengippi Village | 1 | 0 | Health (Community Health Assistant; no map produced due to time constraints) |
| 36 | Loima | Lorengippi Village | 1 | 0 | CFW (CHP) |
| 37 | Loima | Lorengippi Village | 0 | 1 | Administration |
| 38 | Turkana Central | Lodwar | 1 | 0 | Veterinary (Agrovet owner) |
| 39 | Turkana Central | Lodwar | 1 | 0 | Environment |

In this study, FCM steps included:

1. **Concept Identification and Map Construct** – Each mapping session was guided by a semi-structured interview that asked participants to identify components (nodes) and relationships (connections) between people, other animals, and the environment, with a focus on health-related processes and outcomes. System boundaries were defined inductively through the participatory modeling process rather than specified *a* priori, allowing concepts and relationships to emerge from participants' own understanding of the SES.

2. **Data Treatment and Homogenization** – We transcribed hand-drawn maps into Mental Modeler (www.mentalmodeler.com) for refinement. Adjacency matrices were exported for further analysis. During this step, terminology was standardized and semantically similar components were merged to reduce network complexity while preserving conceptual meaning.

3. **Transitive Closure and Aggregation** – We applied fuzzy transitive closure to each adjacency matrix using CIET Map 2.2 (https://ciet.org/fcm/) and aggregated individual maps into stakeholder maps by using the FCMapper package in R (Table 2; [32]).

4. **Network Analysis and Visualization** – We imported group maps into Gephi (version 0.10.0; [33]) for network visualization and analysis, including structural characteristics of each map and centrality of individual components. Stakeholder map adjacency matrices are included in S1 Table.

To assess whether the complexity of aggregated stakeholder cognitive maps was influenced by sample size, we conducted a series of Spearman rank correlations between the number of individual maps per stakeholder group and key network metrics (e.g., number of components, number of connections, density, and connections per component). We also fitted a simple linear regression model to examine the relationship between sample size and number of components. Correlation coefficients ($\rho$) and p-values were calculated for each comparison, with significance determined at $p \leq 0.05$. All analyses were performed in R (Posit team, 2024).

Because maps varied in composition, raw centrality values were not directly comparable. Therefore, we calculated normalized z-scores in Microsoft Excel to assess the relative influence of each component within a map. These standardized scores allowed us to compare the importance of shared components across stakeholder groups (Equation 1).

Equation 1. Standardized centrality (z-score) used to compare the relative importance of shared components across fuzzy cognitive maps with different sizes and structures. $X$ = raw centrality score, $\mu$ = mean centrality within each map,

**Table 2. Makeup of each aggregated stakeholder map.**

| Stakeholder Group | Number of Fuzzy Cognitive Maps | Breakdown by Gender and Method |
|---|---|---|
| Community members | 15 | • 8 male group sessions<br>• 7 female groups sessions |
| Community Frontline Workers | 3 | • 2 mixed-gender group sessions<br>• 1 female individual session |
| Administration | 4 | • 4 male individual sessions |
| Health | 2 | • 2 male individual sessions |
| Veterinary | 2 | • 2 male individual sessions |
| Environment | 5 | • 1 mixed-gender group session<br>• 2 female group sessions<br>• 2 male individual session |
| NGO | 4 | • 1 female individual session<br>• 3 male individual sessions |

and $\sigma$ = standard deviation within each map. This transformation normalizes centrality values within each map enabling meaningful cross-group comparisons.

$$z - \text{score} = \frac{X - \mu}{\sigma}$$

**Qualitative analysis.** We used a Straussian grounded theory (GT) approach that included open and axial coding to analyze the causal relationships identified in the FCM process and examine how stakeholders conceptualize OH [34].

Mapping session transcripts were uploaded to Delve for qualitative analysis ([www.delvetool.com](www.delvetool.com)). To systemically analyze these data, we applied the Corbin and Struass paradigm, coding conditions, action-interactions, and consequences. We adapted this paradigm to explicitly examine how socioecological conditions influence action and interactions among people, other animals, and the environment with health- and wellbeing-related consequences ([Fig 2](Fig 2)).

Coding was conducted iteratively, with regular memo-writing to capture emerging insights and to refine categories as analysis progressed. To enhance rigor, coding was conducted by three researchers, with regular cross-checking of coding decisions and emerging categories.

### Ethics statement

Ethical approval for human subjects research was obtained through the Tufts University Human Subject Institutional Review Board (IRB ID 00004210), the University of Nairobi Biosafety, Animal Use and Ethics Committee (REF: FVM BAUEC/2023/447), and the National Commission for Science, Technology, & Innovation (License No: NACOSTI/P/23/30716). Prior to each mapping session, participants were informed about the purpose, procedures, data

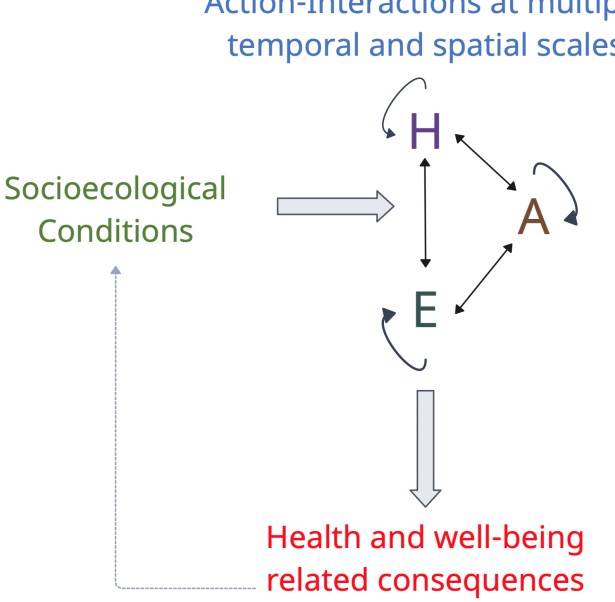

**Fig 2. The One Health coding paradigm (OHCP) includes socioecological conditions, action and interactions at multiple temporal and spatial scales, and health and well-being related consequences.** Action-interactions are coded across humans **(H)**, other animals **(A)**, and the environment **(E)**. Solid black arrows reflect relationships between and among people, domestic and wild animals, and the natural environment. The dashed gray line represents feedbacks and causal relationships through which consequences reconfigure socioecological conditions over time, shaping subsequent actions and interactions in a continuous, dynamic process. Made in Creately ([www.app.creately.com](www.app.creately.com)).

[https://doi.org/10.1371/journal.pone.0338167.g002](https://doi.org/10.1371/journal.pone.0338167.g002)

handling, confidentiality, and the voluntary nature of participation. Verbal informed consent was obtained from all participants. Consent was documented by the research team at the time of data collection and witnessed by EFG or CK, in accordance with the approved Tufts IRB study protocol.

## Results

### Structural characteristics of stakeholder maps

The structural characteristics of stakeholder maps varied in both size and complexity (Table 3). The number of maps per group ranged from 2 (health and veterinary) to 14 (community). The community map contained the highest number of components (84) and connections (704), while the health map had the fewest. Network density ranged from 0.063 (environment) to 0.17 (veterinary). Community and NGO maps had the highest number of connections per component – 8.4 and 7.5, respectively. The distribution of node types also varied: the administrative and CFW maps had the highest proportion of driver nodes, while the community map had no receiver nodes. Most stakeholder maps were composed primarily of transmitter and ordinary nodes, with relatively few receiver-only concepts across all groups.

Results of Spearman rank correlations between the number of individual maps aggregated per stakeholder group and structural characteristics showed strong, statistically significant correlations between sample size and both the number of components ($\rho = 0.98$, $p < 0.001$) and the number of connections ($\rho = 0.87$, $p = 0.01$) (Table 4). In contrast, we found no significant association between sample size and either network density ($\rho = -0.65$, $p = 0.11$) or average connections per component ($\rho = 0.44$, $p = 0.33$) (Table 4). A simple linear regression supported these findings, showing a positive but non-significant relationship between sample size and number of components ($\beta = 1.97$, $p = 0.12$), likely due to the limited number of stakeholder groups (n = 7).

### Component-level comparison across stakeholder groups

The heatmap and dendrogram based on Jaccard similarity indices (J) illustrates patterns of conceptual overlap across stakeholder group maps (Fig 3). The environment group was the most distinct, while health and NGO groups shared the greatest similarity. Overall overlap was low (J = 0.12–0.30).

**Table 3. Structural characteristics for each aggregated stakeholder map. Driver variables have a positive outdegree and zero indegree, ordinary variables have both positive outdegree and indegree, and receiver variables have a zero outdegree and a positive indegree [35].**

| Structural characteristics | Admin-istration | Community | CFWs | Environment | Health | Veterinary | NGO |
|---|---|---|---|---|---|---|---|
| Maps (n) | 4 | 14 | 3 | 5 | 2 | 2 | 4 |
| # components | 72 | 84 | 48 | 83 | 37 | 40 | 68 |
| # connections | 354 | 704 | 221 | 431 | 190 | 271 | 510 |
| Density | 0.069 | 0.10 | 0.098 | 0.063 | 0.14 | 0.17 | 0.11 |
| Connections per component | 4.9 | 8.4 | 4.6 | 5.2 | 5.1 | 6.8 | 7.5 |
| Number of driver/receiver/ordinary | 43/3/25 | 40/0/44 | 25/0/23 | 34/3/45 | 12/2/23 | 13/2/25 | 20/1/47 |

**Table 4. Spearman's rank correlation to assess the relationship between sample size and structural characteristics across stakeholder maps. A p-value ≤ 0.05 is considered statistically significant.**

| Metric | Spearman's ρ | p-value |
|---|---|---|
| # components | 0.98 | **<0.001** |
| # connections | 0.87 | **0.010** |
| Density | −0.65 | 0.111 |
| Connections/component | 0.44 | 0.328 |

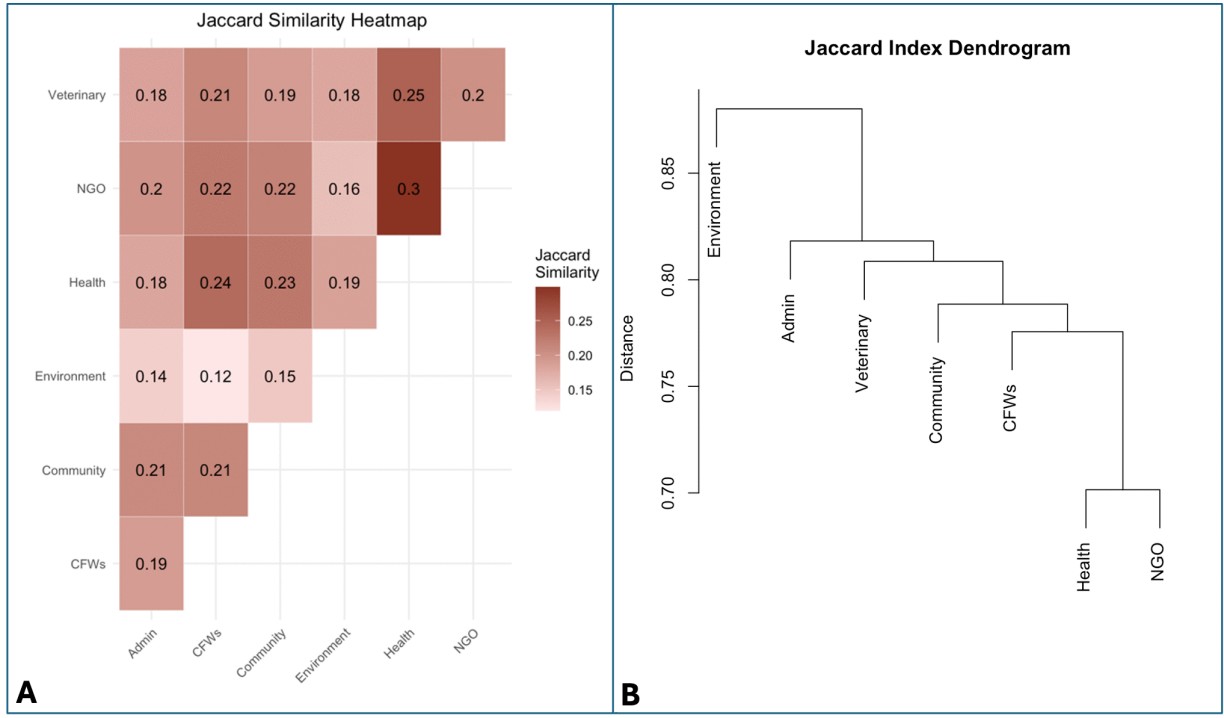

**Fig 3. Panel A: Jaccard indices heatmap ranging from 0 (none) to 1 (perfect) conceptual overlap. The Jaccard similarity index is calculated as a/(a+b+c) where a is the number of shared components, and b and c are the number of components unique to each map. Panel B: Dendrogram derived from the Jaccard-distance matrix (1-J). Branch height indicates the maximum pairwise distance between clusters at each node.**

Analysis of the top ten components ranked by raw centrality (Table 5) and standardized z-scores for components shared across four or more maps (Fig 4) reveal both convergence and divergence across stakeholder groups. Human health, livestock health, and nutrition consistently appeared in the top three components across groups (Table 5), yet comparison of z-scores show substantial variation in the degree of emphasis (Fig 4).

**Table 5. Top ten components ranked by centrality across stakeholder groups. ASF=animal source food, CADR=community animal disease reporters.**

| Rank | Admin | Community | CFWs | Environment | Health | Veterinary | NGO |
|------|-------|-----------|------|-------------|--------|------------|-----|
| 1 | Livestock health | Human health | Human health | Livestock health | Human health | Human health | Human health |
| 2 | Insecurity | Livestock health | Nutrition | Human health | Human resources | Livestock health | Nutrition |
| 3 | Human health | Nutrition | ASFs | Nutrition | Livestock health | ASFs | Livestock health |
| 4 | Veterinary services | Health services | Livestock health | Water resources | Water resources | Nutrition | Insecurity |
| 5 | Nutrition | ASFs | Water resources | Water quality | Water infra-structure | CADRs | Water resources |
| 6 | Pasture resources | Veterinary services | Drought | Environ-mental degradation | ASFs | Private sector | Pasture resources |
| 7 | Water resources | Water resources | Livestock Health | Crop production | Insecurity | Drought | Health services |
| 8 | ASFs | Income | Veterinary services | Deforest-ation | Health services | Public sector | Migration |
| 9 | Income | Pasture resources | Pasture resources | Pasture resources | WASH | Referral | Veterinary services |
| 10 | Health services | Foodstuffs | Foodstuffs | ASFs | Disease surveillance | Training/kits | Kimormor |

**Fig 4. Heatmap of standardized centrality scores (z-scores) for components included in four or more stakeholder group maps.** Stakeholder groups included administration (e.g., chiefs, village administrators, community, community frontline workers (CFWs), human health (e.g., public health officer), environment (e.g., environment, water, and climate change officials), NGOs (e.g., TUPADO, USAID Nawiri, and IRC), and veterinary (e.g., veterinary officers). Higher scores (green) indicate greater relative importance within each group's map, while lower scores (red) indicate lower relative importance. The "Max-Min Difference" column reflects the range across groups for each component, highlighting the most variation in perceived importance.

| Component | Admin | Community | CFWs | Environment | Health | NGO | Veterinary | Max-Min Difference |
|---|---|---|---|---|---|---|---|---|
| Insecurity | 3.67 | 0.57 | 0.31 | -0.40 | 0.90 | 2.21 | -0.59 | 4.26 |
| Livestock health | 3.93 | 3.15 | 1.25 | 4.95 | 1.98 | 2.39 | 2.55 | 3.70 |
| Veterinary services | 2.39 | 2.34 | 0.90 | | -0.46 | 0.96 | -0.45 | 2.85 |
| Water resources | 1.56 | 1.91 | 1.08 | 2.54 | 0.98 | 1.78 | -0.15 | 2.69 |
| Human resources | -0.69 | -0.25 | | | 2.00 | 0.10 | | 2.69 |
| Crop agriculture | -0.72 | -0.53 | | 1.44 | | 0.24 | -1.20 | 2.64 |
| Water quality | -0.28 | -0.46 | -0.72 | 1.60 | 0.09 | -0.94 | | 2.54 |
| ASFs | 1.45 | 2.42 | 1.78 | 0.79 | 0.97 | 0.02 | 2.13 | 2.40 |
| Health services | 1.08 | 2.48 | 0.08 | | 0.88 | 1.18 | | 2.39 |
| Foodstuffs | -0.42 | 1.67 | 0.41 | | | -0.46 | | 2.12 |
| Human health | 3.09 | 3.49 | 4.06 | 4.45 | 3.42 | 4.51 | 2.59 | 1.92 |
| Drought | 0.08 | 1.48 | 1.06 | 0.19 | -0.40 | 0.79 | 0.58 | 1.88 |
| Water infrastructure | -0.38 | 0.19 | | 0.44 | 0.98 | -0.05 | -0.66 | 1.64 |
| Pasture resources | 1.70 | 1.69 | 0.45 | 0.82 | 0.14 | 1.71 | 0.33 | 1.57 |
| Vaccination | | | | -0.34 | -1.04 | 0.52 | -0.66 | 1.56 |
| Kimormor | 0.05 | -0.34 | -0.10 | -0.67 | -0.16 | 0.81 | | 1.48 |
| WASH | -0.31 | -0.66 | 0.33 | -0.34 | 0.82 | 0.51 | | 1.48 |
| Education | -0.57 | -0.23 | -0.36 | 0.68 | -0.64 | -0.77 | 0.07 | 1.45 |
| Nutrition | 1.74 | 2.97 | 3.18 | 2.67 | | 2.45 | 2.06 | 1.44 |
| Migration | 0.07 | 0.55 | -0.38 | | | 1.05 | -0.22 | 1.43 |
| Gender inequity | | 0.59 | -0.67 | | -0.78 | -0.63 | | 1.37 |
| Distance | | 0.67 | -0.69 | | | -0.55 | 0.05 | 1.35 |
| Referral | -0.63 | -0.54 | -0.72 | | | | 0.55 | 1.27 |
| Zoonotic disease | -0.29 | -0.12 | -0.41 | -0.38 | -0.34 | -0.12 | -0.89 | 0.77 |
| Prosopis | -0.04 | -0.38 | | -0.15 | -0.66 | | | 0.62 |
| Climate change | | | | -0.31 | -0.48 | 0.04 | -0.29 | 0.52 |
| Wet season | -0.07 | 0.09 | -0.31 | -0.12 | | | | 0.40 |

Livestock health, for example, has the second largest z-score difference, with environment, administration, and community groups assigning it the highest importance (Fig 4). Water resources also consistently appeared in the top ten components across groups (Table 5) and was highly prioritized except for the veterinary group (Fig 4).

Group-specific priorities also emerged. Insecurity was prioritized by the administrative and NGO groups (Table 5) but assigned lower importance by environment and veterinary groups (Fig 4). Health and veterinary groups also highlighted technical elements such as disease surveillance, referral, and training (Table 5). The environment group diverged the most strongly, emphasizing ecosystem-related concepts such as environmental degradation and deforestation (Table 5), while assigning relatively higher importance to crop agriculture and water quality compared to other groups (Fig 4).

Veterinary services showed the third largest z-score difference, with low priority among the health and veterinary groups (Fig 4). This likely reflects the inclusion of private sector and public sector as unique components in the veterinary group map that diluted the saliency of "veterinary services" as a shared component across groups in the comparative analysis.

To visually illustrate the structure and content discussed above, Fig 5 presents the Environment stakeholder group fuzzy cognitive map. This group was the most conceptually distinct, showing that *environment degradation*, driven by *poverty* and *deforestation*, has a strong negative impact on human and livestock heath. Water competition and contamination because of people and livestock sharing limited water sources featured prominently in environmental actors' narratives, reflected by the negative relationships between *livestock population*, *water resources*, and *water quality.* Upstream drivers like *ecosystem-friendly policies* are positioned as potential mitigators of these pressures. Overall, the emphasis on degradation of natural resources underscores how environmental actors frame OH primarily through an ecological lens.

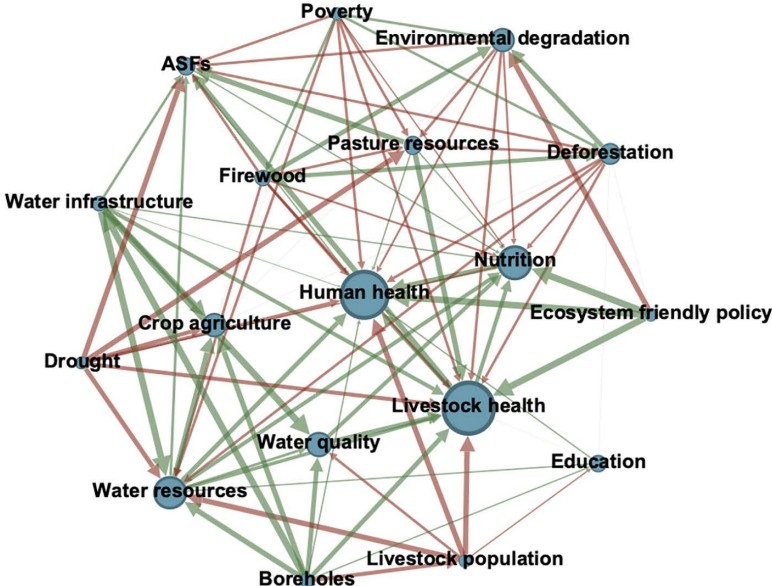

**Fig 5. Environmental stakeholder group fuzzy cognitive map.** Node size represents degree centrality. Green and red arrows represent positive and negative relationships, respectively. Arrow width corresponds to edge weight. Only nodes with centrality > 7 and edges > 0.5 are displayed for clarity of visualization. ASF = animal source foods (e.g., milk and meat). Created in Gephi (version 0.10.0).

In comparison, the Health stakeholder group fuzzy cognitive map reveals a distinctly different framing, centered on technical and disease-related dimensions of OH (Fig 6). For example, there is a strong positive relationship between *human resources* (e.g., community health promoters, subcounty officers, and health facility personnel), *disease surveillance*, and *human health*. Furthermore, Health stakeholders emphasized *behavior change* and what they termed *ignorance* as key determinants of health outcomes. This aligns with their focus on health education, further reflected in the strong positive relationship between *community-led total sanitation* (CLTS) and *water quality. Zoonotic disease* had a medium negative relationship with *human health*, and a weaker negative relationship with *livestock health,* illustrating the focus on people over livestock in this group.

Overall, the FCM findings demonstrate that stakeholder share several high-level priorities yet differ markedly in how they interpret cause and effect within the OH system. These patterns point to differences in experience and framing, which we explore in greater detail through the grounded theory analysis in the next section.

## Grounded theory analysis

In the following section, we highlight areas of divergence among stakeholder groups based on the GT analysis.

### 1. Deforestation, degradation, and human impacts on ecosystems

Environmental stakeholders differed from other groups by emphasizing the direct role of human activities on environmental outcomes. While most participants linked resource availability to rainfall patterns and structural factors such as water infrastructure, environmental actors highlighted human-driven drivers such as deforestation, mining, and pollution—alongside restoration measures like reforestation and grass reseeding.

Deforestation was a central concern. For example, a participant in the Environment directorate emphasized: *"Deforestation is becoming a menace in Kenya. They* [the community] *cut trees for fuel and charcoal"* (MS 39). Several linked

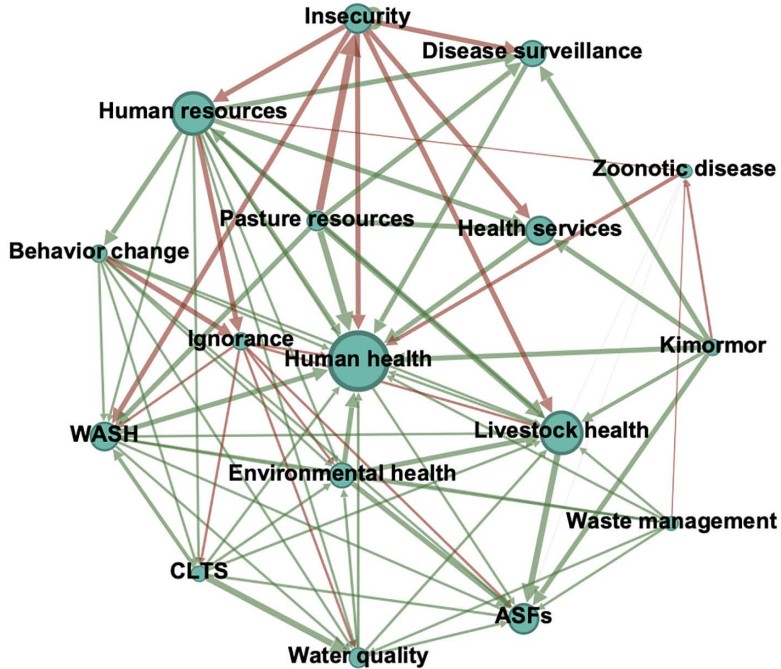

**Fig 6. Health stakeholder group fuzzy cognitive map.** Node size represents degree centrality. Green and red arrows represent positive and negative relationships, respectively. Arrow width corresponds to edge weight. Only nodes with centrality >5 and edges >0.4 are included for clarity of visualization. CLTS = community-led total sanitation; WASH = water, sanitation, and hygiene. Created in Gephi (version 0.10.0).

tree-cutting with the refugee camp in Kakuma. Another highlighted the expansion of urban areas as a driver: *"People are cutting trees to increase the number of houses, and it has a negative impact on the environment"* (MS 27).

An official in the Climate Change Directorate described a cascade of consequences from deforestation, explaining: *"With deforestation comes land degradation. Once the area is bare, the wind comes, and rain even washes away the soils. The vulnerability of people increases"* (MS 29). The same participant also linked these factors to a loss of biodiversity and less rain – claiming that trees *"attract rain clouds."*

Environmental stakeholders also highlighted mining and oil extraction as a major driver of degradation, noting *"mining degrades the land by creating pits,"* and adding, *"oil and gas causes pollution and fine particulate matter which is dangerous to health"* (MS 39). Plastic pollution also featured heavily among environmental stakeholders, with one describing how livestock get obstructed when eating garbage in town. Plastic (and liquid) waste was linked to insufficient waste management systems and infrastructure.

The same actors highlighted reforestation and grass reseeding as important interventions to reduce degradation and the loss of important indigenous trees. For example, an official in the Environment directorate described, *"We bring topsoil, and we add value by planting trees. We even plant some grass"* (MS 39).

Overall, environmental stakeholders emphasized how human activities, such as pollution and resource extraction, directly influence the environment, while other stakeholders primarily focused on rainfall as the main drivers of water and pasture availability, with little attention to anthropomorphic impacts.

### 2. Traditional medical practices in a pluralistic health system

Community members consistently emphasized the importance of traditional medicine for treating both people and livestock, while service providers rarely acknowledge these practices or viewed them as a barrier to formal treatment. One

woman explained, *"Here, there is no hospital. We pick some shrubs/herbs and use them as medicine. Some* [people] *recovery and some die"* (MS 03).

In some instances, traditional practices were seen as very effective, with one participant describing, *"If the child has measles… we slaughter the animal and pour blood on the child, the child will heal completely. White meat is rolled like a cap on the head, and the child is healed"* (MS 12).

For livestock, traditional treatments include branding with a hot stone or iron (e.g., for Contagious Caprine Pleuropneumonia, CCPP) or using traditional remedies like *aloe vera*. Livestock owners described these practices as part of a triage approach: *"The traditional medicine is like first aid as you are looking for drugs"* (MS 18).

Community animal disease reporters (CADRs) who bridge formal and informal systems, provided important nuance. One noted, *"Traditional medicine helps to reduce the severity but does not fully eradicate the infection. It subsides as we wait for the medication and prescription from the vet officers"* (MS 05).

Perceptions of effectiveness varied. While some valued traditional methods, many participants highlighted the greater efficacy of conventional treatments. As one man said about livestock care, *"the drug from the store helps a lot and is more effective. The stone, I'm not sure. Some get better or might die,"* (MS 19) while another stated, *"veterinary services help a lot more than traditional medicine* (MS 15).

Crucially, these narratives did not present traditional and formal care as mutually exclusive. Instead, community members described a pluralistic health system: *"We use traditional ways of treating ourselves using roots from identified trees, but we also acquire medication from the nearby health facilities"* (MS 14). Traditional medicine was valued for its accessibility and perceived benefits, while conventional healthcare and veterinary services were generally seen as more effective, but less accessible.

Despite the prominence of traditional medicine in community accounts, it was almost entirely absent from service provider narratives. Only one non-community stakeholder (a village administrator) discussed traditional practices and framed them as a barrier during an ongoing measles outbreak: "*They believe that measles is not treatable, it's not curable. The moment you take someone to the hospital, he dies"* (MS 22). As a result, *"when they suspect it's measles, they don't reveal it to the community health promoters."* He added, *"They believe in their way of treating it by slaughtering the sheep, smearing the blood. They believe when they do that the person gets healed."*

This divergence highlights how certain culturally embedded practices remain invisible to formal OH actors, potentially limiting the effectiveness of synergistic and complementary practices.

3. **Dietary diversity and wild foods**

Stakeholders differed in how they understood and valued dietary diversity, with institutional actors emphasizing specific agricultural interventions and education, while community members highlighted the importance of sorghum production and wild foods. Acute malnutrition emerged as a distinct concern for frontline health workers and NGOs. Community health promoters described monitoring and referring malnourished children for treatment, including the use of "*Plumpy'Nut*" (MS 04). They also emphasized the importance of a balanced and diverse diet. As one explained, *"children need protein, carbohydrates, and vitamins. The mother is supposed to be trained on how to provide a proper diet… to add vegetables, fruits, some healthy fats, so that the child is healthy"* (MS 36).

An NGO official who was not from Turkana linked cultural norms to dietary diversity challenges, noting: *"Cultural norms contribute to acute malnutrition in this county. People don't know issues like food diversification"* (MS 33). They added that changing these norms can be very difficult: *"It's very hard to convince pastoralists to engage in farming… because traditionally, culturally, they just know to be pastoralists."*

However, community members themselves provided examples of diversification, such as farming sorghum (*ngimwa*) as a supplemental food source to livestock products. They also highlighted the critical role of wild foods, particularly during the dry season when livestock products are scarce. One participant described, *"We eat wild fruits like Elamach, Edapal,*

*Ngakalalio when there is a drought"* (MS 03). In contrast, other stakeholders did not mention wild foods as part of the diet or as an important coping strategy.

Health and NGO actors instead often promoted crop farming and kitchen gardens to improve nutritional outcomes. One Chief explained: *"We have been trained on farming and kitchen gardening* [by an NGO] *to supplement the milk and meat. We are practicing some crop farming and growing watermelons"* (MS 37). A community health assistant echoed this viewpoint, described kitchen gardens as a way to *"help with other nutrients, the greens, the balanced diet"* (MS 35).

Not all stakeholders supported this emphasis on diversification, however. An agrovet owner from the community critiqued the push for diversification, stating: *"When people take most of the animal products, we don't see any nutritional disorders. And I think that is God's plan because we don't see cases of malnourished children…we encourage people to take that protein* (MS 38).

Overall, these narratives reveal clear differences in how dietary diversity is understood and valued across stakeholders, with community members emphasizing crop production where ecologically feasible and wild foods, and institutional actors prioritizing diversification through specific types of agricultural interventions and nutrition education.

4.   **Contrasting perspectives on water quality: Infrastructure vs. behavior**

Water quality was a concern shared across stakeholder group, but community members (especially women) framed the issue in terms of inadequate infrastructure and limited access, whereas public-health oriented actors emphasized hygiene, sanitation, and behavior change. As one community member explained, when the shallow wells dry up during the dry season, *"only the water pan remains to serve the community which is not clean for consumption"* (MS 05). Members of a village water committee (environment group) similarly highlighted infrastructure, suggesting that more boreholes or separating human and animal water points could reduce contamination and improve water quality (MS 06).

In contrast, public-health aligned stakeholders more often attributed poor water quality to hygiene and sanitation practices. An environmental office stated: *"they* [the community] *need to know the best practices of maintaining their own health… keeping the environment clean, especially keeping the water clean"* (MS 39). A health official similarly observed: *"They don't practice good hygiene. They wash there, the animals drink there. So, as a result, people are susceptible to disease"* (MS 25).

Poor sanitation practices, particularly open defecation, were cited frequently by non-community stakeholders. *"Turkana County has one of the highest open defecation rates in Kenya,"* noted an officer in the Water Directorate, adding that this contaminates shallow wells and contributes to cholera outbreaks (MS 28). Diarrhea was also framed differently: health, NGO, and environmental stakeholders linked it to waterborne diseases (e.g., cholera) and poor sanitation and hygiene practices. Community members primarily discussed it as a food safety issue – resulting from consuming meat from animals that have died from illness.

Proposed interventions reflected these perspectives. Health officials emphasized community-led total sanitation (CTLS) and behavior change through hygiene education. For example, one subcounty public health officer noted, *"We try to impart knowledge for them to be responsible about their health… We have some community health promotors digging a pit latrine"* (MS 25). Community members, in contrast, emphasized the importance of water infrastructure development, including boreholes and water pans for livestock.

NGOs often bridged the two approaches, combining investments in infrastructure with hygiene promotion. As one official explained, *"Water infrastructure is key for WASH. We support this community through water infrastructure… We also train the community on hygiene promotion"* (MS 30).

Overall, community actors focused on access and infrastructure, while health and environmental stakeholders prioritized sanitation and hygiene, revealing distinct but complementary understanding of water-related risks and prevention strategies.

5. **The economics of service delivery and access to care**

While access to services was consistently prioritized across community and service provider groups, community members described access to both livestock drugs and referral health services as contingent on the ability to generate cash income, most often through selling goats. *"When the children get sick, we take and sell one of the goats to take the child to the hospital,"* explained one woman (MS 17). Another emphasized, *"You have to sell your animal so that you can get money to travel,"* (MS 20) while another added, *"We buy drugs from the agrovet after selling one of the goats"* (MS 17). CADRs echoed this view, emphasizing how falling livestock prices during droughts limits household ability to pay for care: *"The value of the animal reduces... The amount is small to cater for our needs. This one we can only sell for 600 Ksh"* (MS 05). Community members also blamed traders for driving down prices, with one participant stating, *"Due to brokers, we sell livestock at low prices that aren't enough to buy drugs…"* (MS 08).

Despite the centrality of this strategy in community narratives, only one human health participant mentioned livestock sales as a means of financing care, highlighting a significant disconnect between community realities and service provides' perspectives. Service providers instead emphasized programmatic approaches to improving service delivery, without directly engaging with economic barriers shaping access to care. This tendency may reflect a prevailing perception among health and veterinary actors that communities are unwilling to pay for services. Yet when asked directly, one woman emphasized, *"We are ready to buy* [livestock drugs]. *We love our livestock. We don't have a store here in Lorengippi"* (MS 20). As a result of this narrative, market-based access constraints remained largely absent from formal service planning, even though they are a key determinant of service update for pastoralist households.

## Discussion

Our findings provide empirical evidence that while stakeholders in Turkana broadly agreed on the importance of core domains such as human and livestock health, nutrition, and water resources, they conceptualize how health emerges within a SES in fundamentally different ways. These differences reflect institutional mandates, knowledge systems, and lived experiences. As a result, OH operationalization is constrained not only by structural or resource constraints, but by deeper epistemic and conceptual misalignments that shape problem framing, intervention priorities, and whose knowledge is treated as authoritative.

To interpret these differences, we used a Struassian GT approach and the OHCP [34]. Qualitative approaches remain underrepresented in OH research. Existing studies have largely relied on thematic analysis to explore disease risk, governance challenges, or barriers to implementation [36,37]. Grounded theory methods have been used sparingly, for example, to inform indicator selection for the Global One Health Index [38], develop a OH policy framework [25], and examine pastoralist decision-making [39].

In this study, the OHCP enabled systematic comparison of how stakeholder groups construct causal relationships, revealing the epistemic foundations underlying divergent conceptualizations and priorities. By beginning from the stakeholders' own categories, explanations, and lived experiences, this analysis adopts an emic perspective, foregrounding how participants themselves understand the health dynamics within the SES [40].

While system boundaries were defined inductively by study participants here, the OHCP may also be used in future GT studies to examine more tightly bounded socioecological health challenges, such as zoonotic disease transmission or antimicrobial resistance. The integration of FCM and GT thus supports transdisciplinary knowledge production in OH to inform more equitable, context-sensitive interventions and policies [41].

Our study also introduces an innovative approach to quantifying stakeholder perceptions using standardized centrality z-scores. While other FCM studies identify the relative importance of components among group maps, for example, the rank or number of times a shared component is mentioned [35,42], our use of z-scores enables direct, quantitative comparison across stakeholder groups, accounting for differences in map structure and content. This represents a methodological advancement in FCM that can enhance cross-group analysis and support more rigorous applications in policy and decision-making. In the following section, we explore the implications of our findings for OH practitioners and OH operationalization more broadly.

## Implications for One Health research, policy, and practice

In Turkana, these findings have direct relevance for OH governance and coordination. The limited conceptual overlap observed across stakeholder maps suggests that existing OH mechanisms must reconcile divergent framings and epistemologies rather than simply coordinate actors. The distinct conceptualization of OH among environmental stakeholders, alongside the more similar disease-focused framing shared by human health and veterinary actors, underscores how institutional mandates, language, and training shape how health is understood, contributing to the persistent marginalization of environmental perspectives in OH policy, tools, and practice [5,9,43,44].

At the same time, shared priorities among stakeholder groups offer entry points for integration but require expanding OH beyond its typical biomedical focus. Water resources, for example, can serve as a unifying domain that links health, ecosystems, and livelihoods, but only if approached more broadly through an ecosystem service lens that reflects the lived experiences of community members [45]. Too often, however, it has been framed narrowly through a disease-centric perspective (e.g., waterborne disease or AMR) [46,47]. Reimaging OH through a SES perspective can help to achieve this broader vision [48,49].

Beyond the clear divergence of environmental stakeholders, we also found that agreement on the general importance of components does not necessarily translate into equal prioritization. For example, livestock health consistently ranked among the top priorities across groups, yet standardized centrality scores revealed wide variation in its perceived importance. Thus, negotiating the relative weight of shared priorities is essential in designing OH initiatives, particularly because OH problems embed trade-offs and objective conflicts that require structured negotiation and incentive alignment [50]. For example, in integrated service delivery initiatives such as *Kimormor,* tensions over how to allocate resources across human, livestock, and environmental health services are not just logistical, they reflect power and institutional hierarchy. As a result, *Kimormor* is skewed towards human health services, followed by veterinary services, with little consideration for environmental health [29]. The injection of political and financial support for OH often intensifies sectoral power struggles, with human health actors typically dominating resource flows and shaping operationalization in ways that align with their interests [51,52]. Without explicit mechanisms to ensure more balanced allocations, OH initiatives risk reproducing these silos, splintering collaboration and undermining the systemic benefits of integration. Centering community priorities during the design phase offers one pathway to avoid these pitfalls and aligns with the core OH principle of socio-political and multicultural parity [6].

Our results align with other studies emphasizing the importance of traditional medical practices among the Turkana, even as devolution has expanded access to formal services [25,53–56]. Previous research emphasizes the importance of engaging *Emurons* (seers/traditional healers) for effective program implementation [25,57], yet few studies in the region have explored the potential synergies between traditional and biomedical systems, or assess the effectiveness of traditional medical practices among the Turkana and surrounding pastoralist communities [58]. This highlights the need for researchers and institutional actors in the health and veterinary sectors to engage with traditional medicine not as an "alternative" system, but a complementary source of diagnostic and therapeutic knowledge [59,60]. Our findings challenge the dominant biomedical framing of OH, which privileges scientific expertise while undervaluing Indigenous knowledge, limiting both equality and effectiveness [10,61]. Addressing this imbalance requires recognition of Indigenous knowledge holders as equal, and essential partners in OH initiatives, moving towards *epistemic pluralism* [6,62].

Wild foods also emerged in our study as a critical, yet underrecognized, component of the SES. Their role in nutrition and interactions with other system components such as conflict, illustrates how biodiversity operates as a foundational socioecological determinant, rather than merely an environmental indicator [63]. Recognizing these linkages highlights the need to embed conservation directly within OH policy and practice for not only wild fruits but also medicinal plants [45,64–68]. This perspective redefines conservation not as a competing agenda, but as a prerequisite for sustained well-being.

Conversely, NGO and environment stakeholders prioritized crop production, framing climate-smart agriculture (CSA) and farming more broadly as key pathways to resilience [69], unlike community members who emphasized

livestock-based production. Yet CSA initiatives can have the opposite effect by fragmenting grazing lands or imposing input-heavy practices poorly suited to variable rainfall in the drylands [70]. Pastoralism itself is inherently resilient, developed through millennia of coping with uncertainty and variable rainfall in dryland ecosystems [71,72]. This suggests that externally driven resilience frameworks may erode rather than enhance local adaptive capacity. As the IPCC's recent inclusion of "responses" in its risk framework recognizes [73], intervention must anticipate unintended consequences, especially where climate actions risk undermining long-standing and well-adapted livelihoods like pastoralism in Turkana.

Pastoralists already integrate human, animal, and environmental linkages through traditional ecological knowledge (TEK), such as detailed observations of plant and soil conditions that guide grazing decisions [74]. This type of TEK has been successfully combined with technical rangeland assessments to monitor degradation, highlighting how transdisciplinary collaboration can support knowledge integration [75,76]. Engaging communities in the planning, implementation, and refinement of OH initiatives and policy can help ground OH approaches in contextually relevant and locally informed practices, progressing toward functional, interactive, and ultimately self-mobilization forms of participation [19].

### Study limitations

Our study had several limitations. First, the small sample size influenced network analysis results. We found a positive correlation between aggregated map sample size and both the number of components and the number of connections (Table 4). This suggests we did not reach full concept saturation – defined as the point at which additional mapping sessions no longer yield new concepts. Sarmiento, Dion [77] recommend 12–15 individual maps per group to ensure new concept saturation, a threshold only meet for the Community group in this study. Our relatively broad system boundary, while effective in capturing how diverse stakeholders conceptualized OH, likely contributed to this result [78].

Importantly, neither network density nor average connections per component were significantly associated with sample size. This suggests that while larger samples increase the number of components up to a point, key structural metrics may remain relatively stable. Thus, density and related structural metrics may provide a more reliable basis for cross-group comparisons when sample sizes are limited.

Second, variation in how participants defined and organized concepts influenced centrality scores. For example, the veterinary group distinguished between "public veterinary services" and "private veterinary services," which lowered the centrality of "veterinary services" relative to other groups. This highlights how variation in concept granularity and homogenization can affect quantitative comparisons and underscores the need for careful and transparent aggregation during FCM analysis [79]. Although we standardized terminology where appropriate, variation in language and framing reflects the inherent complexity of participatory systems research and the pluralistic perspectives central to OH.

Finally, grouping participants by job title or role did not always capture deeper differences in knowledge, experience, and priorities. For example, GT analysis revealed a clear divide between service providers and service users, and between participants from Turkana verses those from outside the region, that cut across formal stakeholder categories. The environment group was especially heterogeneous, which may explain some of the observed conceptual variation.

Despite these limitations, our integrated FCM–GT approach provides a transferable framework for identify stakeholder conceptualization of OH in complex SES. Future research should test this framework in other contexts to examine how different governance structures and cultural settings shape OH priorities and conceptualization.

### Conclusion

Our integrated FCM-GT approach reveals that barriers to OH operationalization are shaped by differences in knowledge, experience, and problem framing across sectors and society within a pastoral SES. This insight reframes OH from simply a coordination mechanism into a deliberative space for negotiating knowledge, power, and priorities. To translate this into practice, OH must:

1. Elevate environmental and Indigenous knowledge actors from peripheral consultation to co-leadership.

2. Promote *epistemic pluralism*, legitimizing multiple ways of knowing, including TEK as sources of evidence.

3. Explicitly and systematically identify how stakeholders conceptualize OH, using participatory systems approaches and social research to inform more effective OH implementation [80,81].

Our approach offers a transferable framework for participatory OH research and contributes actionable insights for more inclusive and ecologically grounded OH implementation. Ultimately, understanding how diverse stakeholders conceptualize OH relationships can transform it from a multidisciplinary aspiration into a truly transdisciplinary practice capable of addressing the complex health and sustainability challenges of our time.

## Supporting information

**S1 Table. Aggregated stakeholder group adjacency matrices.** Each worksheet in the Excel file represents an aggregated fuzzy cognitive map adjacency matrix for a specific stakeholder group. Values represent the weighted directional relationship between components, with weights ranging from −1 (strong negative relationship) to +1 (strong positive relationship).
(XLSX)

## Acknowledgments

The authors sincerely thank Dr. Daniel Esimit Echakan, County Deputy Director for Preventive and Promotive Health; Dr. Benson Etelej Long'or, County Director of Veterinary Services; and Ms. Phoebe Ekali, County Director of Environment, for their invaluable support in making this work possible. We are also grateful to the County Chief Officers and County Executive Committee Members from the relevant ministries, as well as the sub-county administrators from both the national and county governments, for their support.

We thank the chiefs and village administrators for their critical role in community mobilization and logistics, and to the sub-county technical officers in health, veterinary, and environment for their assistance throughout the research process.

Finally, our deepest gratitude goes to the community members and study participants, whose time, insights, and lived experiences form the foundation of this research. Your contributions are deeply appreciated.

## Author contributions

**Conceptualization:** Evan F. Griffith, Janetrix Hellen Amuguni.

**Data curation:** Evan F. Griffith, Angela Opondoh, Jonah Levin.

**Formal analysis:** Evan F. Griffith, Angela Opondoh, Jonah Levin.

**Funding acquisition:** Janetrix Hellen Amuguni.

**Investigation:** Evan F. Griffith, Angela Opondoh, Catherine Kaluwa, Erenius Lochede Nakadio, Job Ronoh Kipkemoi.

**Methodology:** Evan F. Griffith, Angela Opondoh, Catherine Kaluwa, Janetrix Hellen Amuguni.

**Project administration:** Catherine Kaluwa, Erenius Lochede Nakadio, Kipkorir Rotich, Job Ronoh Kipkemoi, Jacob Mutua, Janetrix Hellen Amuguni.

**Resources:** Janetrix Hellen Amuguni.

**Supervision:** Catherine Kaluwa, Janetrix Hellen Amuguni.

**Visualization:** Evan F. Griffith.

**Writing – original draft:** Evan F. Griffith.

**Writing – review & editing:** Evan F. Griffith, Angela Opondoh, Catherine Kaluwa, Erenius Lochede Nakadio, Kipkorir Rotich, Job Ronoh Kipkemoi, Jonah Levin, Jacob Mutua, Siobhan M. Mor, Janetrix Hellen Amuguni.

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
