## [Decision Letter · Decision Letter 0]

29 Jan 2026

Dear Dr. Griffith,

Thank you for submitting your manuscript to PLOS ONE. After careful consideration, we feel that it has merit but does not fully meet PLOS ONE’s publication criteria as it currently stands. Therefore, we invite you to submit a revised version of the manuscript that addresses the points raised during the review process.

Kindly have a look at the comments from the reviewers and address them accordingly.

We look forward to receiving your revised manuscript.

Kind regards,

Eric Ogola, MPH

Academic Editor

PLOS One

Journal Requirements:

2. In the ethics statement in the Methods, you have specified that verbal consent was obtained. Please provide additional details regarding how this consent was documented and witnessed, and state whether this was approved by the IRB.

“This research was generously funded by a Cummings Foundation grant (V710458) to H.J.A. Website: https://www.cummingsfoundation.org/.”

6. We note that Figure 1 in your submission contain map images which may be copyrighted. All PLOS content is published under the Creative Commons Attribution License (CC BY 4.0), which means that the manuscript, images, and Supporting Information files will be freely available online, and any third party is permitted to access, download, copy, distribute, and use these materials in any way, even commercially, with proper attribution. For these reasons, we cannot publish previously copyrighted maps or satellite images created using proprietary data, such as Google software (Google Maps, Street View, and Earth). For more information, see our copyright guidelines: http://journals.plos.org/plosone/s/licenses-and-copyright.

7. Please remove your figures from within your manuscript file, leaving only the individual TIFF/EPS image files, uploaded separately. These will be automatically included in the reviewers’ PDF.

8. We note you have included a table to which you do not refer in the text of your manuscript. Please ensure that you refer to Table 5 in your text; if accepted, production will need this reference to link the reader to the Table.

9. Please include a copy of Table 6 which you refer to in your text on page 19.

Reviewers' comments:

Reviewer's Responses to Questions

**Comments to the Author**

1. Is the manuscript technically sound, and do the data support the conclusions?

Reviewer #1: Yes

Reviewer #2: Yes

2. Has the statistical analysis been performed appropriately and rigorously?

Reviewer #1: Yes

Reviewer #2: Yes

3. Have the authors made all data underlying the findings in their manuscript fully available?

Reviewer #1: No

Reviewer #2: No

4. Is the manuscript presented in an intelligible fashion and written in standard English?

Reviewer #1: Yes

Reviewer #2: Yes

Reviewer #1: The manuscript presents an innovative approach combining fuzzy cognitive mapping and grounded theory to examine stakeholder conceptualizations of One Health. Integration of participatory system modeling with qualitative analysis was excellent and well aligned with the study objectives. Additionally, the manuscript is well written and makes a meaningful contribution. However, minor changes are needed.

1. The proposed theory is a valuable framework, but its novelty should be explicitly added. A short description would strengthen the manuscript.

2. There is a variation in the sample size. Could you please add some discussion how may small size affect the stability? of this study during comparison.

3. Add the limitations of this study more explicitly.

4. The manuscript is detailed and well organized, but reducing some repetition would increase the readability

5.Figures and Tables.

Some figures (e.g., FCM visualizations) are dense and may be difficult for readers unfamiliar with FCM to interpret. Consider adding brief explanatory captions or references to supplementary material.

Table numbering appears inconsistent in places (e.g., Table 5 vs. Table 6); please check carefully.I have not found table table 6 in the mansucript.

6.

Reviewer #2: The study uses a validated instrument (Smartphone Addiction Scale – Short Version) to measure smartphone overuse and applies binary logistic regression to explore associations between overuse and self-reported physical and mental health outcomes. The sample size is adequate, and the statistical analyses are properly conducted.

It would be helpful to expand slightly on the application of logistic regression—specifically, whether assumptions such as multicollinearity and goodness-of-fit were tested. Although the use of this method is appropriate, adding a short note on diagnostics would enhance transparency.

Given the cross-sectional design, I recommend to avoid wording that implies causality (e.g., "smartphone use caused"). Instead, consider using terms like “associated with” or “linked to.”

According to PLOS ONE policies, data underlying the findings should be openly accessible unless restricted. You currently state data are available upon request. Please consider depositing an anonymised version of the dataset in a recognised public repository to comply with open data practices.

While the SAS-SV is clearly described, the manuscript would benefit from clarification on how physical (e.g., neck or back pain) and mental health outcomes (e.g., anxiety, depression) were assessed—i.e., whether validated instruments were used or whether these were self-reported perceptions.

**Do you want your identity to be public for this peer review?** For information about this choice, including consent withdrawal, please see our Privacy Policy

Reviewer #1: No

Reviewer #2: No

---

## [Author Response · Author response to Decision Letter 1]

11 Feb 2026

Manuscript ID: ONE-D-25-61857

Title: Stakeholder priorities and conceptualization of One Health: Insights from fuzzy cognitive mapping and grounded theory

Dear Dr. Ogola,

We sincerely thank you and the reviewer for your careful evaluation of our manuscript and for the constructive feedback. We have carefully addressed all comments and believe the manuscript has been strengthened as a result.

Below, we provide a point-by-point response to your comments and Reviewer #1. As noted below, Reviewer #2’s comments appear to refer to a different manuscript.

Academic Editor and Journal Requirements

1. Formatting Requirements

We have revised the manuscript to ensure full compliance with PLOS ONE style and formatting guidelines.

2. Clarification of Verbal Consent Documentation

We have updated the manuscript with the following changes (Page 14, lines 246-251): “Prior to each mapping session, participants were informed about the purpose, procedures, data handling, confidentiality, and the voluntary nature of participation. Verbal informed consent was obtained from all participants. Consent was documented by the research team at the time of data collection and witnessed by EFG or CK, in accordance with the approved Tufts IRB study protocol.”

3. Funding Information in Manuscript

All funding information has been removed from the manuscript body and retained only in the Funding Statement section of the online submission form.

4. Grant Information Consistency

We reviewed and confirmed that grant information is now consistent across all sections of the submission.

5. Role of the Funder Statement

We included the following statement in the cover letter: “This research was funded by a Cummings Foundation grant (V710458) awarded to H.J.A. The funders had no role in study design, data collection and analysis, decision to publish, or preparation of the manuscript.”

6. Copyrighted Map Image (Figure 1)

We removed the original copyrighted basemap and replaced Figure 1 with a version that complies with CC BY 4.0 licensing requirements.

7. Removal of Figures from Manuscript File

All figures have been removed from the manuscript file and uploaded separately as individual TIFF files. Figure legends remain in the manuscript.

8–9. Table Numbering Issues

Table numbering has been corrected throughout the manuscript. Table 5 starting on page 16 is now references in the text. There is no Table 6.

10–11. Reference List Review

The reviewer comments did not include recommendations to cite specific works.

Reviewer #1

Comment 1: Clarify novelty of proposed theory (OHGT)

We expanded the Discussion to more explicitly articulate the novelty and broader applicability of One Health Grounded Theory (OHGT), clarifying its theoretical contribution beyond the immediate case study (Page 28, lines 543-559).

Comment 2–3: Sample size variation and limitations

We revised the Study Limitations section to explicitly address how sample size variation may affect concept saturation, centrality analysis, and structural stability across stakeholder maps (Pages 33-34, lines 660-688).

Comment 4: Reduce repetition

Thank you for this comment. Without more specific details regarding what repetition the reviewer is referring to it was hard to know what to change. However, we have made the following revisions with the intention of reducing repetition:

1) We deleted the last two paragraphs of the introduction which included information that was repeated in the Methods section (e.g., sampling and participant recruitment).

2) We deleted the “Study Site Selection,” as this presents information that is included in other parts of the Methods and Discussion section.

3) Removed any reference to the ethics process (Page 8) as this is included in the “Ethics Statement” (Page 12).

4) Finally, we edited the discussion to remove any references to results.

Comment 5: Clarify FCM figures

We added a concise overview of FCM at the beginning of the Methods section and ensured that figure legends clearly explain interpretation of node size, edge color, and arrow thickness. Adjacency matrices are included as supplementary material.

Reviewer #2

The comments provided under Reviewer #2 refer to a study involving the Smartphone Addiction Scale (SAS-SV) and logistic regression analysis, which is unrelated to our manuscript. We respectfully believe this review was inadvertently assigned from another submission.

Data Availability

Adjacency matrices for each stakeholder group are provided in the supplementary material. Qualitative transcripts are not publicly available due to participant confidentiality and IRB restrictions; however, coding structures and analytical procedures are described in detail in the Methods.

Conclusion

We greatly appreciate the constructive feedback provided by the Academic Editor and Reviewer #1. The revisions have strengthened the manuscript’s clarity, transparency, and theoretical contribution. We believe the revised version more clearly articulates its empirical and methodological contributions to participatory One Health research.

Sincerely,

Evan Griffith, DVM, MPH, MS

---

## [Editor Report · Decision Letter 1]

16 Feb 2026

Stakeholder priorities and conceptualization of One Health: Insights from fuzzy cognitive mapping and grounded theory

PONE-D-25-61857R1

Dear Dr. Griffith,

We are pleased to inform you that your manuscript has been judged scientifically suitable for publication and will be formally accepted for publication once it meets all outstanding technical requirements.

Within one week, you'll receive an e-mail detailing the required amendments. When these have been addressed, you’ll receive a formal acceptance letter and your manuscript will be scheduled for publication.

Kind regards,

Eric Ogola, MPH

Academic Editor

PLOS One
---

## [Editor Report · Acceptance letter]

PONE-D-25-61857R1

PLOS One

Dear Dr. Griffith,

I'm pleased to inform you that your manuscript has been deemed suitable for publication in PLOS One. Congratulations! Your manuscript is now being handed over to our production team.

Kind regards,

on behalf of

Dr. Eric Ogola

Academic Editor

PLOS One